# The Volatile Profile of Brussels Sprouts (*Brassica oleracea* Var. *gemmifera*) as Affected by Pulsed Electric Fields in Comparison to Other Pretreatments, Selected to Steer (Bio)Chemical Reactions

**DOI:** 10.3390/foods11182892

**Published:** 2022-09-17

**Authors:** Sophie M. Delbaere, Tom Bernaerts, Mirte Vangrunderbeek, Flore Vancoillie, Marc E. Hendrickx, Tara Grauwet, Ann M. Van Loey

**Affiliations:** Laboratory of Food Technology, Department of Microbial and Molecular Systems (M2S), KU Leuven, Kasteelpark Arenberg 22, PB 2457, 3001 Leuven, Belgium

**Keywords:** pulsed electric fields, electroporation, Brussels sprouts, enzymatic reactions, pretreatment, fingerprinting, multivariate data analysis, flavor

## Abstract

Pulsed electric fields (PEF) at low field strength is considered a non-thermal technique allowing membrane permeabilization in plant-based tissue, hence possibly impacting biochemical conversions and the concomitant volatile profile. Detailed studies on the impact of PEF at low field strength on biochemical conversions in plant-based matrices are scarce but urgently needed to provide the necessary scientific basis allowing to open a potential promising field of applications. As a first objective, the effect of PEF and other treatments that aim to steer biochemical conversions on the volatile profile of Brussels sprouts was compared in this study. As a second objective, the effect of varying PEF conditions on the volatile profile of Brussels sprouts was elucidated. Volatile fingerprinting was used to deduce whether and which (bio)chemical reactions had occurred. Surprisingly, PEF at 1.01 kV/cm and 2.7 kJ/kg prior to heating was assumed not to have caused significant membrane permeabilization since similar volatiles were observed in the case of only heating, as opposed to mixing. A PEF treatment with an electrical field strength of 3.00 kV/cm led to a significantly higher formation of certain enzymatic reaction products, being more pronounced when combined with an energy input of 27.7 kJ/kg, implying that these PEF conditions could induce substantial membrane permeabilization. The results of this study can be utilized to steer enzymatic conversions towards an intended volatile profile of Brussels sprouts by applying PEF.

## 1. Introduction

Cruciferous vegetables belonging to *Brassicaceae*, including Brussels sprouts, broccoli, cauliflower, rapeseed, cabbage, mustard, and kohlrabi, are known to exhibit bitter and sulfurous flavor notes which can determine the degree of consumption by humans, indicating the importance of this sensorial quality characteristic [1,2,3,4,5,6,7,8,9,10,11]. The existence of flavor notes in these vegetables witnesses (bio)chemical reactions (including enzymatic and non-enzymatic reactions) in which important quality-related substrates and enzymes, intrinsically present in the vegetables, play a major role. 

The most prominent enzymatic reaction pathways imparting the flavor in Brussels sprouts are the enzymatic conversion of sulfur-containing health-promoting phytochemicals, namely glucosinolates (GSLs) (which are β-thioglucoside-N-hydroxysulfates and abundantly present in *Brassica* vegetables) and the oxidation of polyunsaturated fatty acids (PUFAs) catalyzed by myrosinase (MYR) (β-thioglucosidase, EC 3.2.1–3.2.3) and lipoxygenase (LOX) (non-heme iron-containing dioxygenase, EC 1.13.11.12) with associated enzymes (i.e., hydroperoxide lyase (HPL) (EC 4.2.99.-), alcohol dehydrogenase (ADH) (EC 1.1.1.1) and alcohol acetyltransferase (AAT) (EC 2.3.1.84)), respectively [9,12,13,14,15,16,17,18]. Regarding the GSLs–MYR pathway, (sulfurous) (flavor-affecting) compounds such as isothiocyanates (ITCs), thiocyanates and nitriles can be formed resulting from the enzymatic hydrolysis of GSLs [1,3,4,19,20,21,22,23,24]. The GSLs that are reported in the literature to be a common source of bitter notes are sinigrin, progoitrin and gluconapin which also largely exist in Brussels sprouts [2,3,10,24,25]. The hydrolysis is initiated by a D-glucose cleavage that forms an unstable thiohydroximate-*O*-sulfonate from which it can generate the aforementioned reaction products and is regulated by several extrinsic factors (e.g., temperature, pH, presence of cofactors (e.g., epithiospecifier protein), identity of the GSLs) [8,9,11,15,18,22,26,27,28,29]. Moreover, (enzyme-catalyzed) breakdown of S-methyl-L-cysteine sulfoxides by cysteine (sulfoxide) lyase (C-S lyase) (EC 4.4.1.10) can contribute to the formation of sulfur-containing flavor-affecting compounds in Brussels sprouts [30]. As for the PUFAs–LOX pathway, oxidation of PUFAs generates (C6 and C9) aldehydes and alcohols which are associated with ‘green-like’ flavors [17,25,31]. In order to enable enzyme–substrate interactions, membrane disruption (e.g., by chopping, chewing or mixing) needs to occur since compartmentalization (separated by cell membranes) is hindering the interaction [9,11]. More specifically, MYR is located in specialized myrosin cells in the plant tissue of *Brassica* plants whereas GSLs are located in GSLs-accumulating cells [32]. Other quality-related enzymes such as C-S lyase, LOX, HPL, ADH, and AAT, are mainly present in the plant cell cytoplasm while corresponding substrates are existing in the plant cell vacuole [19,33]. 

Besides enzymatic reactivities, non-enzymatic reactions also play a role in determining the resulting flavor of Brussels sprouts and other *Brassica* vegetables [19,29,34]. The latter reactions can generate (i) products due to thermal degradation of GSLs and PUFAs; (ii) products resulting from Maillard reactions and the successive side reactions; and (iii) products that arise by (thermally induced) autoxidation [12,18,19,25,29,35,36,37]. 

Hence, it is clear that during processing of *Brassica* vegetables, both the level of tissue disruption as well as the thermal load during processing may impact and, thus, may be applied to steer the volatile profile of *Brassica* vegetables [6,7,36,37,38,39,40]. 

An emerging technology in food processing is pulsed electric fields (PEF). This technique involves delivering a number of high electric voltage (0.5–20 kV/cm) short pulses (µs–ms) to (a(n) (intact) biological) material placed in a specified treatment chamber (batch or continuous) filled with a conductive medium and in between two electrodes. This technique can cause an increase in the transmembrane potential which causes reversible or irreversible permeabilization of the cell membrane (pore formation), known as electroporation [41,42,43,44,45,46]. The effectiveness/degree of permeabilization depends, on the one hand, on process parameters (e.g., electrical field strength, pulse width and shape, frequency, specific energy input, treatment time, and temperature) and on the other hand, on the characteristics of the food to be treated and medium (e.g., dimensions of the (plant) cell, conductivity, pH, ionic strength) [42,44,47,48,49]. These parameters are interrelated to each other which makes the understanding of the influence of one parameter challenging [42]. Electrical field strengths of 12–20 kV/cm, and even up to 35 kV/cm for liquid foods, are usually applied to inactivate microbial cells (cell size 1–10 µm) and enzymes for shelf life extension/preservation purposes [42,43,45]. Lower field intensities (in the range of 0.1–4 kV/cm) are usually applied to affect plant cells (cell size 40–200 µm) [42,43,49,50]. At these lower electrical field strengths, PEF has already been demonstrated to be a useful technique for the improvement in the release of valuable compounds, to enhance extraction and for drying and cutting effectiveness [41,43,44,45,47]. It is speculated that PEF can induce cell membrane permeabilization which will eventually result in the release of intracellular material that affects the biochemical reactions in (vegetable) products [42,44,48,51,52,53,54]. The effect of a PEF treatment on the (residual) enzyme activities in food products has already been extensively studied. However, most studies mainly focused on the implementation of higher electrical field strengths that cause the inactivation of enzymes [50]. At lower electrical field strengths, the effect of PEF on the volatile profile of foods has only been studied for a limited number of matrices, such as onion [55]. To the best of our knowledge, no studies have focused on the effect of PEF at low-electrical-field strengths on the volatile profile of *Brassica* vegetables. Hence, it seems promising to consider this technique in current study, since PEF is hypothesized to enable intrinsic (bio)chemical reactivities (to a certain extent), imparting the presence of volatile compounds by disrupting a tissue-based system only partially, as opposed to a mix step which aims to disrupt the tissue more extensively and a heat step which will inactivate enzymes instantly.

In this context, the influence of PEF at low-electrical-field strengths and other pretreatments that aim to steer enzyme–substrate reactivities on the volatile profile of Brussels sprouts will be further explored and compared in this study. This ‘targeted steering’ approach is, to the best of our knowledge, not the main focus in previous studies that have already been published. The first part will focus on PEF (in combination with heat) at specified conditions compared to other pretreatments, selected in the context of regulating (bio)chemical reactivities. Combinations of mixing (disruption) and heating are implemented (in different orders of sequence). Moreover, insight into the underlying reactivities that occurred during treatment will be elucidated. The second part will specifically focus on the impact of different selected PEF conditions, with exclusion of heat, on the volatile profile as a result of membrane permeabilization.

## 2. Materials and Methods

### 2.1. Implementing PEF versus Other Pretreatments to Compare the Volatile Profiles

#### 2.1.1. Raw Material 

Raw Brussels sprouts (*Brassica oleracea* var. *gemmifera*) were purchased on the day of harvesting. To verify batch variability, two batches, cv. Cryptus, were acquired in the second and the third weeks of January 2021 and originated from Staden, Belgium, and Houthulst, Belgium. Brussels sprouts with diameters of 15–25 mm were selected given their industrial relevance and convenience. Until processing, the vegetables were stored in a refrigerator at 3 °C for a maximum of 5 days. 

#### 2.1.2. No Pretreatment (NoPT) (Control)

Brussels sprouts were washed with tap water to remove the remaining soil before tapping dry with some paper, and very small, dirty, or damaged pieces were discarded. The Brussels sprouts were put into low density polyethylene bags and were frozen in liquid nitrogen. On the day of analysis, frozen Brussels sprouts were grinded using a Grindomix GM200 (Retsch GmbH, Haan, Germany) and subsequently mixed in a closed Thermomix (Vorwerk, Wuppertal, Germany) with saturated NaCl solution (1:1 (*w*:*v*)) which enables inhibition of enzyme activities (verified beforehand) without triggering heat-induced changes.

#### 2.1.3. Mixing Followed by Heating (Mix + Heat)

Brussels sprouts were cleaned as described in Section 2.1.2 and were mixed in a closed Thermomix with demineralized water (1:1 (*w*:*v*)) for 1 min at 10,700 rpm. Subsequently, the system was set to rest for 1 h at room temperature to allow enzymatic conversions. Afterwards, enzymatic conversions were stopped by heating the disintegrated system for 10 min at 95 °C while gently stirring. The latter conditions were determined with a qualitative peroxidase (POD) test according to Adebooye et al. (2008) ensuring POD negative activity, which is considered one of the most heat-stable enzymes in vegetables [56,57]. The vegetable system was then cooled in a cooling room at 3 °C. The above steps were repeated until all Brussels sprouts were treated. Treated Brussels sprouts were pooled afterwards and were then divided in a cooling room (3 °C) into new 50 mL transparent polyethylene terephthalate tubes with a polyethylene cap. Filled tubes were frozen in liquid nitrogen and stored in a freezer at −40 °C until analysis. 

#### 2.1.4. Heating Followed by Mixing (Heat + Mix)

Washed Brussels sprouts were packed in a single layer in low-density polyethylene bags and subjected to a heat step in a water bath (Memmert WTB50, Gent, Belgium) for 15 min at 95 °C to inactivate the endogenous enzymes (ensuring POD negative activity). Subsequently, the bags were cooled down in an ice bath for at least 10 min after which all Brussels sprouts were pooled. Pooled heat-treated Brussels sprouts and demineralized water were put together in a closed Thermomix (1:1 (*w*:*v*)) and were mixed at the highest speed frequency (10,700 rpm) for 1 min. These steps were repeated until all vegetables were treated. Subsequent pooling, tube filling and freezing until analysis were similar to the conditions described in Section 2.1.3. 

#### 2.1.5. Pulsed Electric Fields (PEF) Followed by Heating and Mixing (PEF + Heat + Mix)

The PEF treatments were performed with a PEF unit in batch configuration (Cellcrack III, Elea-DIL, German Institute for Food Technologies, Quakenbrück, Germany). This unit was equipped with a medium-sized treatment chamber consisting of two parallel stainless-steel electrodes for which the width, height, and thickness amounted 20.0, 20.5, and 0.5 cm, respectively. The interelectrode distance amounted to 29.7 cm and the volume of the treatment chamber was 12.2 L. Standardized tap water was prepared by adding 1.3376 g NaCl and 0.2006 g CaCl_2_.H_2_O to 5 L of ultrapure Milli-Q water and was used as a treatment medium (conductivity of approximately 600 µS/cm at 21 °C). A total amount of 650 g Brussels sprouts was inserted into the treatment chamber to which standardized tap water was added until a total mass of 5 kg was achieved. A fabricated lid in the same material as the treatment chamber was used to ensure submersion of the Brussels sprouts. The vegetables were then submitted to 30 monopolar pulses of 30 kV resulting in an electrical field strength of 1.01 kV/cm. The number of pulses was based on preliminary experiments, in which the conductivity of the surrounding medium in which PEF-treated Brussels sprouts were placed was measured with a Testo^®^240 conductivity meter with cell type 07 mS (Testo, Lenzkirch, Germany) after treating Brussels sprouts with different numbers of pulses. The number of pulses was selected at which no additional significant increase in the conductivity was observed (data not shown). The energy input per pulse, the specific energy input per pulse and the (total) specific energy input amounted 450 J/pulse, 90 J/kg·pulse, and 2.7 kJ/kg, respectively. The latter was calculated according to Equation (1) [54].
(1)$$\mathrm{Total}\;\mathrm{specific}\;\mathrm{energy}\;\mathrm{input}=\frac{V^{2}C\;n}{2m}$$
where V (V) is the voltage, C (F) is the capacitance of the energy storage capacitor, n is the number of pulses, and m (kg) is the mass of the sample. 

The pulse width was 225 ± 19 µs which was acquired using an online digital oscilloscope (Tektronix, Köln, Germany). 

After PEF treatment, incubation for 1 h at room temperature was performed to allow enzymatic reactions. Afterwards, the exact same heat (to inactivate enzymes), mixing and subsequent storage steps were followed as described in Section 2.1.4. 

### 2.2. Impact of Parameter Variation in the PEF Treatment

#### 2.2.1. Raw Material

Brussels sprouts were purchased at a local supermarket (Leuven, Belgium) for which diameters amounted 25–30 mm. Pre-handling of the Brussels sprouts was similar to the conditions described in Section 2.1.1. All treatments were carried out within one week of purchase.

#### 2.2.2. PEF + Incubation

The PEF treatment was similar to the conditions described in Section 2.1.5. However, to investigate the impact of PEF conditions on the volatile profile of Brussels sprouts as a result of membrane permeabilization, both the electrical field strength and the (total) specific energy input varied ranging from 1.01 to 3.00 kV/cm and 2.7 to 27.7 kJ/kg, respectively. All parameter values for the PEF treatments in the medium sized and small sized treatment chambers are presented in Table 1. Parameter values were chosen in order to enable comparison of samples that only differ in one parameter. In addition, a non-PEF-treated sample was taken (control). 

After each of the PEF treatments, Brussels sprouts were removed from the standardized medium. PEF-treated Brussels sprouts and control samples (i.e., non-PEF-treated Brussels sprouts) were packed into low density polyethylene bags and subjected to an incubation step in a water bath at 40 °C for 1 h, intended to maximize enzymatic reactivities. The temperature during incubation was based on preliminary tests which showed to clearly cause enzymatic conversions on a disrupted system (data not shown). After 1 h, the bags were cooled for at least 10 min in an ice bath. Thereafter, instead of using a heat step, enzymatic reactivities were inhibited by mixing the Brussels sprouts with saturated NaCl solution (1:1 (*w*:*v*)) to avoid heat-induced chemical conversions. The material was subsequently placed into 50 mL falcon tubes, frozen in liquid nitrogen and stored in a freezer at −40 °C until analysis. 

### 2.3. Analysis of the Volatile Profile

#### 2.3.1. Sample Preparation

The processed samples were thawed overnight in a cooling room at 3 °C. 0.8 g of sample was placed into a 10 mL amber glass vial (VWR International, Radnor, PA, USA) with 3 mL saturated NaCl solution (to increase the concentration of volatile compounds in the headspace [58]) and 0.2 mL demineralized water. The latter amounts were determined based on a saturation test of the fiber using a dilution series while a headspace of 6 mL was maintained (data not shown). The vials were closed using metal screw-caps with a PTFE/silicone septum seal (Grace, Columbia, MD, USA). For each type of treatment, six replicates were analyzed. To each vial, an amount of 100 µL of diluted 3-heptanone solution was added as an internal standard using a gastight syringe in order to be able to detect potential fluctuations in the signal and to monitor the performance of the instrument. Batch variability was observed to be negligible. Therefore, regarding the comparison of different pretreatments (*cfr.* 2.1), combining the data of two batches could be enabled causing 12 replicates per pretreatment.

#### 2.3.2. HS-SPME-GC-MS Fingerprinting 

The headspace–solid phase microextraction–gas chromatography–mass spectrometry (HS-SPME-GC-MS) method was based on the method described by Kebede et al. (2015) [7]. Vials were homogenized and transferred to the cooling tray of the CombiPal autosampler (CTC Analytics AG, Zwingen, Switzerland) which was maintained at 10 °C. Volatile fingerprinting was enabled using a gas chromatographic system (GC 7890B, Agilent Technologies, Santa Clara, CA, USA) coupled with a mass selective detector (MSD) (5977A, Agilent Technologies, Santa Clara, CA, USA). The incubation time was 8 min at 40 °C under agitation at 500 rpm resulting in the transfer of the volatiles to the headspace of the vial. Extraction took 20 min at the same temperature using a specific 30/50 µm divinylbenzene/carboxen/polydimethylsiloxane (DVB/CAR/PDMS) fiber (StableFlex, Supelco, Bellefonte, PA, USA). The latter parameters were optimized beforehand for the Brussels sprouts matrices under study by means of an experimental design aimed at maximizing the number of peaks and the total peak area (data not shown). Fibers were preconditioned beforehand according to the manufacturer instructions. Consequently, the fiber with adsorbed volatiles was inserted into the GC-injection port at 230 °C for 2 min to enable the thermal desorption of the volatiles in a split mode (1:5). The volatile compounds were then separated using an HP Innowax column (Agilent Technologies J&W, Santa Clara, CA, USA) (60 m × 0.25 mm i.d., 250 µm $d_{f}$). Helium (purity ≥ 99.9999%) was used as a carrier gas with a constant flow of 1.273 mL/min and a pressure of 138.13 kPa. A specific oven program was used with a starting temperature of 40 °C for 2 min, followed by heating to 120 °C at 4 °C/min, heating to 200 °C at 7 °C/min, 2 min at 200 °C and heating to 250 °C at 50 °C/min. This resulted in a total running time of 36 min and 26 s per sample. The temperature of the ion source and quadrupole amounted 230 and 150 °C, respectively. The electron ionization (EI) mode at 70 eV in scanning mode (*m*/*z* 35–400) at 3.9 scans/s was used to obtain the mass spectra. All samples to be compared were analyzed in a random order using the same fiber. Fiber degradation, monitored by adding control samples in each sequence, was not observed. 

### 2.4. Multivariate Data Analysis

All obtained chromatograms were deconvoluted by means of the Automated Mass Spectral Deconvolution and Identification System (AMDIS) (Version 2.72, 2014, National Institute of Standards and Technology, Gaithersburg, MD, USA). This software was also used to build a retention index (RI) calibration file for which homologous series of n-alkane standards (C8–C20) were made which were analyzed under the used GC-MS conditions and for data compound identification using the spectral library of NIST software (NIST14, version 2.2, National Institute of Standards and Technology, Gaithersburg, MD, USA) [59,60]. The Mass Profiler Professional software (MPP) (version B12.00, 2012, Agilent Technologies, Diegem, Belgium) allowed peak alignment, peak filtering, and baseline correction resulting in a data table with compiled volatile profile data showing the X-variables in the columns (volatile compounds) and Y-variables in the rows ((pre)treatment(s) (conditions)/classes/groups). Multivariate data analysis (MVDA) was performed using Solo software (Version 8.7.1, 2020 Eigenvector Research, Wenatchee, WA, USA). Firstly, the data were pre-processed by enabling mean-centering and weighing the data by their standard deviation in order to give all data equal variance. Secondly, a Principal Component Analysis (PCA) was conducted as a multivariate technique to explore the data set and to detect potential outliers. Thirdly, Partial Least Square Discriminant Analysis (PLS-DA) was used as a modeling technique which aimed at maximizing the covariance between the categorical Y-variables and the X-variables. The number of latent variables (LVs) that was chosen to build a multivariate model was based on the lowest number that resulted in an optimal class separation. The latter could be elucidated by analyzing the Root Mean Squared Error of Cross Validation (RMSECV). Scores and loadings plots were overlayed and gave biplots which depict how the samples differed from each other based on their volatile profiles allowing qualitative assessment of the differences in volatile profiles of the samples. All biplots were constructed using OriginPro8 (Origin Lab Corporation, Northampton, MA, USA). In a final stage of the data analysis, Variable IDentification coefficients (VID) were assigned to each variable for each class. A VID coefficient is a quantitative measurement representing the correlation between a specific X-variable and the Y-variable(s) as designed by the model and allows to rank the detected volatiles according to their discriminative behavior [12,61]. For each class, volatile compounds with a VID with an absolute value higher than 0.800 were referred to as discriminant volatiles (markers). Discriminant compound plots were plotted depicting the mean peak area of a compound in a specific class as a function of treatment. Confirmation of the identity of the markers was performed by comparing the RI with those found in the available literature. If the RI was not found in the literature or did not match the value described in literature, corresponding compounds were defined as ‘*tentatively identified*’ and ‘*unidentified*’, respectively. Moreover, threshold match and reverse match were set to 80% as a threshold for acceptance of the spectral data. In addition, for a selected set of markers, confirmation was undertaken using analytical standards when available. 

### 2.5. Statistical Analysis

Statistical analyses were conducted using Tukey’s HSD tests in JMP Software (JMP Pro16, SAS Institute Inc., Cary, NC, USA) (p-value of 0.05) to perform significance tests between the mean peak areas of the discriminant volatiles depicted in the discriminant compound plots.

## 3. Results and Discussion

### 3.1. Impacts of PEF and Other Pretreatments on the Volatile Profile of Brussels Sprouts

#### 3.1.1. Qualitative and Quantitative Classification of the Volatile Profiles

Representative total ion chromatograms after each pretreatment can be found in the Appendix A (Figure A1). The volatile profiles of differently processed samples were compared using MVDA. A PLS-DA model was set up and enabled a detection of 118 different volatiles over all chromatograms. The total Y-variance explained by the model, in which 3 LVs were implemented, amounted to 91.74%. 

A graphical summary of the analytical data depicting the differences and relations in volatile profile between the groups being compared can be displayed by means of biplots. The x and y axes on a biplot depict LVs which are acquired after constructing a PLS-DA model based on the analytical data. The three biplots are shown in Figure 1 since the explained Y-variance of each of the LVs in the model was approximately the same. Differently processed samples (classes/groups) are indicated with different symbols. Open circles on the biplots represent the volatile compounds. The closer a volatile is depicted towards a specific group, the more that volatile is representative for that group. Vectors on the biplot represent the correlation loadings pointing to the different classes. The closer the vectors of two groups are positioned to each other, the more comparable their volatile profiles. As observed in Figure 1, NoPT, Mix + Heat, and Heat + Mix pretreatment led to clear separable groups since the vectors point to different directions. A PEF + Heat + Mix pretreatment in the current conditions, however, only was seen to be slightly different based on the third LV compared to the Heat + Mix pretreatment which signifies that the PEF step (in combination with subsequent heating) did not or only slightly impacted the expected (quality-related) enzyme–substrate interactions. It is hypothesized that using more intense PEF conditions could lead to more pronounced permeabilization of membranes enclosing quality-related substrates and enzymes which was probably not achieved under the current conditions. This will be further elaborated in a second part of this study in which possible heat-induced effects will be excluded (Section 3.2). 

In order to quantitatively select volatiles that are responsible for the distinct volatile profiles, the VID procedure was implemented for which the threshold was set at 0.800. This threshold resulted in 57 discriminant components accounting for 79% of the total peak area over all detected volatile components in the headspace of all samples. VID, identity, chemical class and RI of the discriminant components can be found in Table 2.

#### 3.1.2. Interpretation of the Markers in the Headspace after Different Pretreatments on Brussels Sprouts

In this section, possible links to (bio)chemical reaction pathways that could have taken place/were induced during pretreatment are formulated. Both PEF-treating and mixing of the matrix were expected to induce substrate conversions catalyzed by enzymes, either by a partial or an extensive tissue disruption. In all treated samples, a heat step was implemented which also needs to be taken into consideration when relating volatiles to specific pathways. Due to these complex combinations of (bio)chemical reactions that could have occurred during pretreatment, it is challenging to unequivocally ascribe a specific compound to a specific reaction pathway. However, hypothesis-driven links are formulated in the following sections.

##### Sulfurous Compounds, Nitriles, and Isothiocyanates

Specific compound plots of discriminant compounds related to the GSLs–MYR pathway are represented in Figure 2. Dimethyl disulfide, dimethyl trisulfide, and methyl (methylthio)methyl disulfide, observed to be significantly present in the headspace after a Mix + Heat pretreatment (Figure 2), might be derived from the thermal degradation of enzymatically formed reaction products (e.g., sulforaphane) generated by MYR that catalyzed GSLs conversions [67]. In the first step of this pretreatment (i.e., extensive tissue disruption during mixing), enzymatic reaction products might have been formed by the enabled interaction between MYR and GSL due to decompartmentalization, which could be subsequently thermally degraded into these compounds during the heat step [12]. Dimethyl disulfide and dimethyl trisulfide have previously been described as responsible for the typical odor of (cooked) *Brassica* vegetables [2,23,68]. The latter compound was also observed as a marker in thermally treated broccoli puree for which the broccoli was priorly blanched (ensuring POD negative activity), indicating that this compound could also result from solely thermal pathways [7]. Also in the study of Engel et al. (2002), sulfurous compounds (e.g., dimethyl trisulfide) were observed in the volatile profile of priorly blanched cooked chopped cauliflower [2]. Furthermore, sulfurous compounds can be derived from the (enzymatic) decomposition of cysteines substituted with an S-compound as well as their sulfoxides (e.g., S-methyl-L-cysteine sulfoxide) by C-S lyases, which is in agreement with other studies in which dimethyl disulfide is the main reported compound derived from this pathway [27,30,68,69,70]. Additionally, dimethyl disulfide and dimethyl trisulfide were observed to be major and minor products derived from the thermal degradation of sulfur-containing amino acids [67]. Moreover, the presence of dimethyl disulfide can be caused by the oxidation of methanethiol [12].

In contrast to sulfurous compounds, nitriles (e.g., hexanenitrile) were already markedly present in the headspace of the NoPT sample (Figure 2). This might be explained by their presence in the raw material and/or due to the (minor) induction of enzymatic conversions by the initial wounding of the tissue during/after the harvest which could induce enzymatic conversions to a certain extent. 4-(methylthio) butanenitrile and benzyl nitrile are observed to increase after a Mix + Heat pretreatment. Their presence can be related to MYR-catalyzed conversions (if the epithiospecifier protein is still active, which favors the formation of nitriles) by prior extensive tissue disruption which is more plausible compared to the possible non-enzymatic heat-induced formation (e.g., via thermal degradation) since these compounds would otherwise also be higher in abundance in the other heat-treated samples lacking a prior enzyme inductive step.

Notable is that 2-butenenitrile was dominant in headspaces of samples obtained after implementing a heat step on a tissue-based system (i.e., on a PEF-treated material or raw material) (Figure 2). Since no enzyme inductive step was included in the Heat + Mix pretreatment, it seems reasonable that the significantly higher presence of this nitrile in the headspace after this pretreatment is related to thermal degradation of the present substrates (i.e., GSLs), which are hypothesized to be much less present after a Mix + Heat pretreatment given that part of them will have been enzymatically converted into other compounds during the mixing step. As already reported, the sensibility upon this thermally induced degradation reaction also depends on the side structure of the GSL, the presence of Fe^2+^ ions, and the matrix [28]. The finding that nitriles can be related to thermal reactivities, can be additionally verified by the literature since nitriles were reported as dominant reaction products of thermally degraded GSLs, while MYR was already inactivated, in kohlrabi, white and red cabbage by Hanschen et al. (2018), in thermally treated broccoli by Kebede et al. (2013) and in rapeseed oil by Mao et al. (2019) and can therefore be stated as the most plausible explanation for the occurrence of the aforementioned nitrile in this study [28,29,67].

It has been stated in the literature that by short-term heating, the epithiospecifier protein, which favors the formation of nitriles, is inactivated while MYR is still active (due to the higher thermal stability of MYR), resulting in a favored formation of ITCs from GSLs in cut and heat-treated vegetables [18,28]. Therefore, ITC formation could also be presumed in this study since both mixing and heating were implemented. Yet, ITCs in the current study are rarely observed as discriminant compounds over the compared samples, except for 4-isothiocyanato-1-butene, which was discriminant in the headspace after a Mix + Heat treatment (Table 2). This compound as well as allyl ITC (VID > 0.700) are seen to be more abundantly present in the headspace after Mix + Heat as observed in the discriminant compound plots (Figure A2, Appendix A) indicating that ITCs might have also arisen as a consequence of the GSLs–MYR interaction caused by extensive tissue disruption. The fact that over all compounds ITCs are less representative compared to nitriles could mainly be explained by the unstable characteristics of ITCs, which possibly have further reacted during the subsequent heat step of all treatments, for instance into sulfurous compounds (*cfr.* supra) [18].

From Figure 2 and as also noticed in the biplots (Figure 1), the PEF treatment only had a limited effect on the resulting enzyme–substrate interactions possibly due to minimal permeabilization of membranes enclosing enzymes and/or substrates under the current PEF conditions.

##### Alcohols, Aldehydes, Ketones, and Furanic Compounds

Selected discriminant alcohols, aldehydes, ketones, and furanic compounds are depicted in Figure 3. 1-hexanol, 3-hexen-1-ol, (E)-3-hexen-1-ol, 2-hexen-1-ol, and 1-penten-3-ol are significantly present in the volatile profile after a Mix + Heat pretreatment as observed in Figure 3. This can be linked to the extensive tissue disruption as a result of the prior mixing step, which will have induced far-reaching (enzymatic) conversions. More specifically, the presence of the discriminant alcohols might be a result of the conversion of PUFAs by LOX, after which the formed odorless hydroperoxides are further converted to (C6) aldehydes by HPL which are then further converted to (C6) alcohols by ADH [17,71]. Even though pentanal was also present in the volatile profile after a Mix + Heat pretreatment, this aldehyde could be observed particularly in the headspace after a Heat + Mix and PEF + Heat + Mix pretreatment. Since PEF under the current conditions probably had no or only a limited effect on the biochemical reaction pathways, as already stated previously, the occurrence of pentanal most likely results from the subsequent heating step. This is evidenced by the fact that the abundance of pentanal in the volatile profile was approximately the same after the PEF + Heat + Mix and Heat + Mix pretreatment (Figure 3). Hence, pentanal could probably be ascribed to the thermal degradation of present substrates since direct enzyme inactivation was intended in the heat step. This compound was also observed in the headspace of cooked broccoli in a study by Hansen et al. (1997) [25]. The main reason why this compound is observed to a lesser extent in the headspace after an extensive tissue disruptive step (Mix + Heat) is believed to be due to the possible prior enzymatic conversions leading to lower amounts of available substrates to be subjected to thermal degradation to form this product. Particularly noteworthy is the presence of certain aldehydes and ketones in the headspace of the NoPT sample as can be seen in Figure 3. The presence of hexanal and (E)-2-hexenal (and (E)-2-pentenal) might be due to the possible damage to the Brussels sprouts upon harvesting prior to mixing with a saturated NaCl solution, which could already have triggered certain PUFA conversions by LOX and HPL leading to these secondary reaction products [17,71,72]. The reason hexanal is minorly observed in the volatile profile after (PEF + )Heat + Mix might be due to the thermal degradation of this product into other products. In the volatile profile after a Mix + Heat pretreatment, the most plausible reason for the lower amounts of these aforementioned compounds is the possible further conversion of these reactive primary degradation products to other secondary (and tertiary) reaction products (such as 2-alkyl furans) during the extensive mix and heat step. The latter is a plausible reason for the presence of 2-ethyl furan in the headspace after a Mix + Heat pretreatment (Figure 3), which is, as described in the literature by Grebenteuch, Kroh, et al. (2021), a possible tertiary lipid oxidation product derived from 2-hexenal which can be induced by a heat step [71]. Moreover, furans, which can be considered as relatively stable products, can be released from ascorbic acid and can be present due to possible Maillard reactions (sugar with specific amino acids) [7,67]. Furthermore, this compound can also be formed from carotenes and organic acids [71,73]. (E,E)-2,4-heptadienal, which is also known to be a secondary reaction product in the PUFAs–LOX pathway, is a marker with a negative VID in the headspace after the (PEF + )Heat + Mix pretreatment. From Figure 3, it can be observed that its presence is significantly higher after a Mix + Heat pretreatment and in the headspace of the NoPT sample, which might indicate that the LOX conversion of its corresponding fatty acid, namely linolenic acid, is likely to happen both during/after the harvest and after the mix step [71]. During the (PEF + )Heat + Mix pretreatment, this compound might have been thermally degraded while during Mix + Heat, further formation in addition to conversion/degradation of this compound might be a plausible explanation for the observed abundance. Well-known tertiary reaction products in the PUFAs–LOX pathway derived from 2,4-heptadienals are methyl ketones. 1-penten-3-one in the current study is observed to be discriminant in the NoPT sample, indicating that far-reaching conversion along this pathway likely happened during tissue wounding upon harvest and might also have been a determinative factor for the observed abundance of (E,E)-2,4-heptadienal in the headspace of the NoPT sample [71,72].

### 3.2. Impact of PEF Conditions on the Volatile Profile of Brussels Sprouts

#### 3.2.1. Qualitative and Quantitative Classification of the Volatile Profiles

In the first part of the current study, it was clearly demonstrated that a PEF treatment at 1.01 kV/cm and 2.7 kJ/kg, followed by heating, was believed to not or only slightly cause permeabilization of membranes enclosing quality-related substrates and enzymes. Therefore, in this second part, we investigated how altering the electrical field strength and the (total) specific energy input during a PEF treatment would affect the volatile profile of Brussels sprouts. Moreover, to exclude heat-induced effects, enzymes were not inactivated by heat but their activities were inhibited by adding saturated NaCl solution. On the one hand, the electrical field strength was elevated up to 3.00 kV/cm and on the other hand, the specific energy input was raised up to 27.7 kJ/kg. Across all PEF-treated and non-PEF-treated samples, a total of 108 volatile components were detected, of which 32 were discriminant (|VID| ≥ 0.800) (Table 3), comprising 81% of the total peak area of all detected volatile compounds. PLS-DA resulted in a model with 4 LVs, describing 95.12% of the total Y-variability. Each of the LVs explained almost an equal proportion of this variability. It was chosen to depict biplots giving a representative presentation of the differences in volatile profiles between the samples (Figure 4). From the biplots, it could be observed that the different PEF treatments clearly caused distinctive volatile profiles, indicating that different reactivities had occurred. The difference in volatile profiles after PEF treatments with different electrical field strengths is mainly explained by LV2, while the difference in volatile profiles of samples obtained after implementing different specific energy inputs but with same electrical field strengths can mainly be explained by LV3 and LV4, for a specific electrical field strength of 3.00 and 1.01 kV/cm, respectively.

#### 3.2.2. Linking Markers to (Bio)Chemical Reaction Pathways Possibly Occurring during and after PEF Treatment

The possible reaction pathways that led to the selected markers will be elucidated in the following sections. It must be noticed that a one-to-one comparison with the results elucidated in Section 3.1.2 cannot be made since the Brussels sprouts that were used to process the samples did not descend from the same supplier. In addition, the first part of the current study (Section 3.1) implemented a heat shock to inactivate the enzymes, which notably contributed to the resulting volatile profile while in this second part of the study, heat-induced changes were excluded since enzyme activities were inhibited by the addition of a saturated NaCl solution.

##### Nitriles

The mean peak areas of selected discriminant nitriles for each of the treatments are summarized in Figure 5. Notable is the presence of nitriles in the headspace of all samples. In the NoPEF sample, their occurrence is possibly related to their (natural) presence in the raw vegetable (possibly induced by enzymatic conversion due to postharvest tissue wounding or tissue wounding during harvest). For benzyl nitrile, 2-butenenitrile and 3-methyl butenenitrile, the abundance is significantly higher in the headspace of the sample that underwent a PEF treatment under an electrical field strength of 3.00 kV/cm and a specific energy input of 27.7 kJ/kg as observed in the specific compound plots (Figure 5). This observation implies that significant permeabilization of membranes enclosing MYR and GSLs due to PEF treatment under these conditions could have occurred enabling a substrate–enzyme interaction. As described by Puértolas et al. (2012) and Toepfl et al. (2006), the permeabilization of the membranes is dependent on the cell size, among others [42,46]. The smaller the size of the cells, the higher the necessary electrical field strength to be implemented in order to enable electroporation [42]. However, clear conclusions regarding the size of the cells enclosing GSLs and MYR in Brussels sprouts cannot be made since available literature on this topic is scarce. In this context, microscopic visualization of the different cell types in Brussels sprouts would be of relevance for this work. Nevertheless, given the current results, it seems reasonable that (one of) the cells enclosing GSLs and MYR might be relatively small, possibly experiencing improved permeabilization at 3.00 kV/cm. Additionally, an increase in the specific energy input played a determinative role in the resulting volatile profile. By increasing the specific energy input, it is reasonable that a possible heating, also known as Joule heating, of the sample could have occurred [74]. The significant temperature rise in the matrix and surrounding water was verified by an additional test. This could imply that thermal effects might be reasonable. Although the main factors that are reported in literature to be responsible for the effect of Joule heating are the tissue conductivity and the electrical field strength, it might be postulated that in the current study, the increment of the specific energy input might have been a prominent factor that caused the occurrence of this phenomenon [74]. The possible effect on the volatile profile of the probable induced Joule heating at these conditions is hypothesized since the abundance of the aforementioned compounds is not significantly higher compared to their presence after a NoPEF treatment if only the electrical field strength was elevated. Since the temperature did not rise above 40 °C, it could be suggested that the possible Joule heating could have mostly contributed to the susceptibility of cell membranes for electroporation. This probably enabled further intrinsic reactivities, over the possible thermal degradations of substrates and/or reaction products as a consequence of this phenomenon [28,42].

Despite the fact that, for the aforementioned nitriles, no large significant differences were observed between PEF-treated samples in which either the electrical field strength or the specific energy input was elevated during the PEF treatment, it could be postulated that PEF (and definitely at a higher electrical field strength) might have influenced the membrane permeability to a certain extent, but that the (relatively low amount of) formed compounds as a consequence of the PEF treatment reacted during the subsequent incubation. This reasoning can be confirmed for 4-(methylthio) butanenitrile which was significantly more abundant in the headspace after a PEF treatment at 3.00 kV/cm and 2.8 kJ/kg (Figure 5). This implies that only increasing the electrical field strength to 3.00 kV/cm possibly already induced enzymatic conversions due to significant permeabilization at the respective conditions leading to this compound. It was clearly noticed that the abundance of this nitrile significantly decreased when the specific energy input was elevated. This might be related to a possible Joule heating as a consequence of elevating the specific energy input that could have affected the abundance due to possible thermally induced reactions.

The incubation at 40 °C for 1 h is hypothesized to also have played a role in the final abundance of nitriles in the different samples since temperature also can have an effect on the abundance of volatiles [75]. In fact, both formation and conversion of concerned compounds could have occurred in all samples during the incubation, but the extent of it was probably determined by the preceding PEF treatment.

##### Aldehydes and Alcohols

Specific compound plots of relevant discriminant aldehydes and alcohols are presented in Figure 6. As was similarly stated for GSLs and MYR, the PEF treatment at an electrical field strength of 3.00 kV/cm in combination with a specific energy input of 27.7 kJ/kg probably caused a substantial increment in the degree of permeabilization of membranes that enclose PUFAs and/or LOX (and associated enzymes) [42]. This permeabilization might have induced the formation of (Z)-3-hexen-1-ol and (E,Z)-2,6-nonadienal as observed in Figure 6. The latter compounds have been reported as secondary products in the PUFAs–LOX (and associated enzymes) pathway [71]. In contrast, primary reaction products of this pathway (i.e., hexanal, (E)-2-hexenal, (Z)-2-hexenal, and 3-hexenal) were perceived to be significantly decreased when the electrical field strength (and more obvious in combination with an additional elevated specific energy input) was elevated. This observation is believed to be due to the possible (enzymatic) conversion of those products into other products such as the aforementioned alcohol (i.e., (Z)-3-hexen-1-ol). This increase in alcohol formation is not observed for the lower energy input at this higher electrical field strength, which might postulate that the further conversion of the primary aldehydes in this sample might not be formed enzymatically, implying that this combination of conditions possibly did not lead to a significant electroporation of the membranes enclosing PUFAs, LOX, and associated enzymes. Another plausible explanation for the observed significantly higher abundance of (Z)-3-hexen-1-ol using the most intense PEF conditions, is that an additional effect of (short-term) Joule heating, as a consequence of a higher energy input, could have played a role [74]. Namely that the effect of this heat-induced phenomenon (only in combination with a higher electrical field strength) could have determined the abundance of this compound as a consequence of possible thermal degradation of specific compounds into this alcohol. (E,E)-2,4-heptadienal is not observed in the samples that underwent a PEF treatment at a higher specific energy input indicating that the effect of possible Joule heating might have caused this product to have reacted. The presence of this compound in the NoPEF sample, as well as the other compounds depicted in Figure 6, might be related to prior tissue wounding in which substrates and enzymes were (partially) released and/or to their (natural) presence in the raw material. The slight decrease in compounds under PEF conditions at lower electrical field strength and/or at lower specific energy input might most reasonably be attributed to further (non-enzymatic) reactions (for instance during subsequent incubation).

## 4. Conclusions and Future Perspectives

In this paper, the impact of PEF and other pretreatments that aimed to steer enzyme-substrate reactions on the volatile profile of Brussels sprouts was investigated. A PEF treatment on Brussels sprouts at an electrical field strength of 1.01 kV/cm and a specific energy input of 2.7 kJ/kg, hypothesized to enable membrane permeabilization, in combination with a subsequent heat step, resulted in a similar volatile profile compared to solely heating. In these samples, the prominent presence of pentanal and 2-butenenitrile (probably arisen due to thermal degradation of the present substrates) was observed, while little to no volatiles related to enzymatic reactivities could be detected. This suggests that the hypothesized membrane permeabilization was not or only minorly achieved using current PEF conditions. It would be relevant for future work to verify this hypothesis, which was stated based on the outcome of the volatile fraction, by use of microscopic analyses to determine the membrane integrity. Mixing prior to heating, on the contrary, was seen to remarkably induce enzymatic conversions of GSLs by MYR and/or of sulfoxides by C-S lyase, presumably due to the extensive decompartmentalization enabled by mixing. Sulfurous compounds such as dimethyl disulfide and dimethyl trisulfide were observed to exhibit discriminative behavior in the volatile profile of this sample which were believed to be mainly present due to the thermally induced breakdown of enzymatically formed reaction products. The implication that the conditions applied during PEF were believed to be insufficient to cause substantial permeabilization of membranes enclosing quality-related substrates and enzymes, raised the question whether more intense PEF conditions would result in increased membrane permeabilization. Therefore, in a second part of this study, a follow-up experiment was established in which a set of selected PEF conditions was applied including more intense conditions and possible heat effects were excluded by not inactivating enzymes using a latter processing step, but by inhibiting enzymatic activities via adding saturated NaCl solution. From these experiments, it seemed reasonable that elevating the electrical field strength up to 3.00 kV/cm probably led to the hypothesized membrane permeabilization of organelles/cells enclosing certain quality-related substrates and/or enzymes as evidenced from the presence of specific nitriles. Additionally, it was postulated that a specific energy input of 27.7 kJ/kg in combination with the higher electrical field strength of 3.00 kV/cm might have been a determining factor in defining the final volatile profile due to a contribution to increased membrane permeabilization by Joule heating.

In future, it would be valuable to consider whether the observed analytical differences in volatile profiles of different samples being compared are also perceived by the human senses of smell and/or taste by implementing in vivo sensory tests, since specific compounds may or may not comprise flavor-active properties, which depends on their concentration.

## Figures and Tables

**Figure 1 foods-11-02892-f001:**
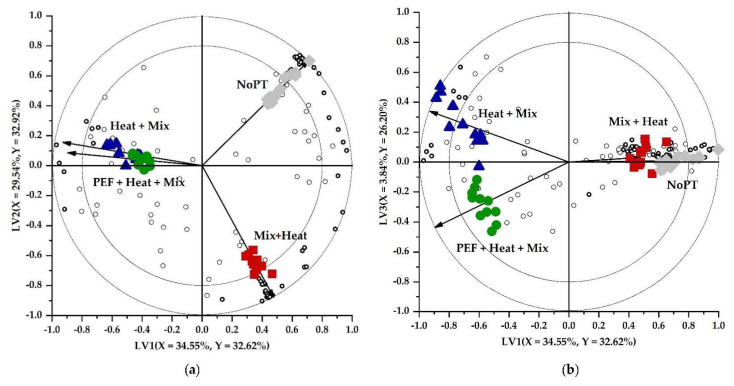
PLS-DA biplots visualizing the effect of pretreatment ((◆) No pretreatment (NoPT), (■) Mix + Heat, (●) PEF + Heat + Mix, and (▲) Heat + Mix) on the volatile profile of Brussels sprouts. Open circles (o) on the biplots represent the headspace components with the discriminant components marked in bold (|VID| ≥ 0.800) (**o**). Vectors depict the correlation loadings for the categorical Y-variables. The variance explained by each LV is indicated in the respective axes. The inner and outer circles depict the correlation coefficients of 0.8 and 1.0, respectively. (**a**) LV2 as a function of LV1; (**b**) LV3 as a function of LV1; (**c**) LV3 as a function of LV2.

**Figure 2 foods-11-02892-f002:**
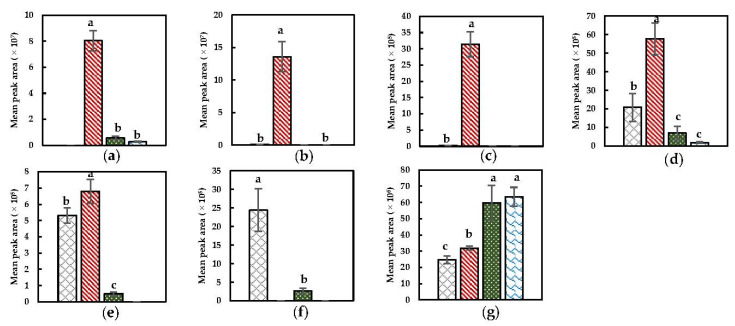
Specific compound plots of selected discriminant sulfurous compound and nitriles in the headspace of differently pretreated Brussels sprouts ((
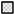
) NoPT, (
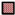
) Mix + Heat, (
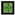
) PEF + Heat + Mix, and (
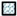
) Heat + Mix). (**a**) Dimethyl disulfide; (**b**) dimethyl trisulfide; (**c**) methyl (methylthio)methyl disulfide; (**d**) 4-(methylthio) butanenitrile; (**e**) benzyl nitrile; (**f**) hexanenitrile; (**g**) 2-butenenitrile. Statistically significant differences are designated by different letters (*p* < 0.05, *n* = 12).

**Figure 3 foods-11-02892-f003:**
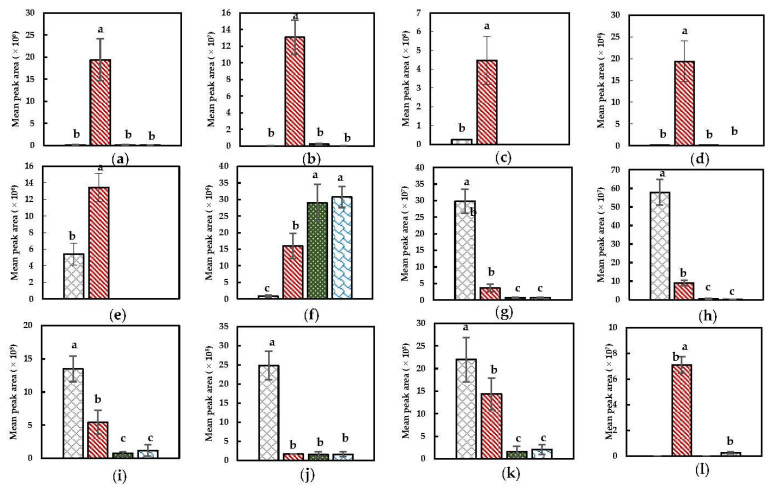
Specific compound plots of selected discriminant aldehydes, alcohols, ketones and furanic compounds in the headspace of differently pretreated Brussels sprouts ((
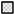
) NoPT, (
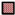
) Mix + Heat, (
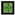
) PEF + Heat + Mix, and (
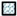
) Heat + Mix). (**a**) 1-hexanol; (**b**) 3-hexen-1-ol; (**c**) (E)-3-hexen-1-ol; (**d**) 2-hexen-1-ol; (**e**) 1-penten-3-ol; (**f**) pentanal; (**g**) hexanal; (**h**) (E)-2-hexenal; (**i**) (E)-2-pentenal; (**j**) 1-penten-3-one; (**k**) (E,E)-2,4-heptadienal; (**l**) 2-ethyl furan. Statistically significant differences are designated by different letters (*p* < 0.05, *n* = 12).

**Figure 4 foods-11-02892-f004:**
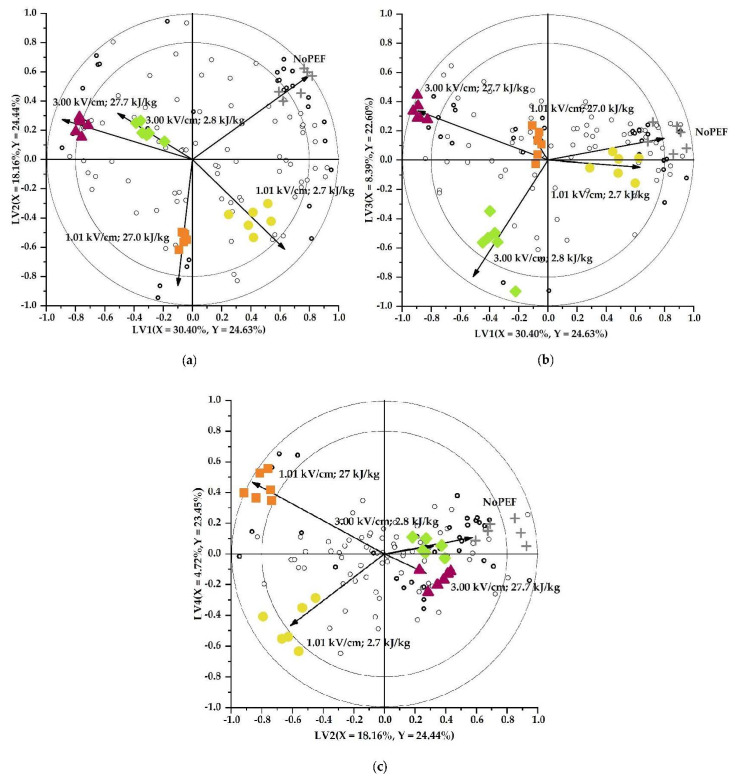
PLS-DA biplots visualizing the effect of different PEF conditions ((**+**) NoPEF, (●) 1.01 kV/cm and 2.7 kJ/kg, (■) 1.01 kV/cm and 27 kJ/kg, (♦) 3.00 kV/cm and 2.8 kJ/kg, and (▲) 3.00 kV/cm and 27.7 kJ/kg) on the volatile profile of Brussels sprouts. Open circles (o) on the biplots represent the headspace components with the discriminant components marked in bold (|VID| ≥ 0.800) (**o**). Vectors depict the correlation loadings for the categorical *Y*-variables. The variance explained by each LV is indicated on the respective axes. The inner and outer circles depict the correlation coefficients of 0.8 and 1.0, respectively. (**a**) LV2 as a function of LV1; (**b**) LV3 as a function of LV1; (**c**) LV4 as a function of LV2.

**Figure 5 foods-11-02892-f005:**
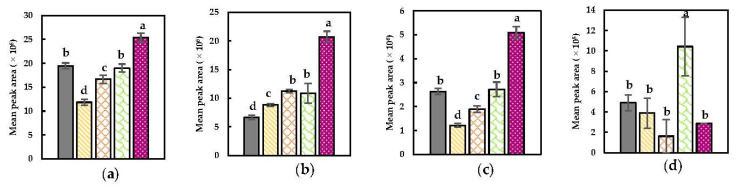
Specific compound plots of selected discriminant nitriles in the headspace of differently PEF-treated Brussels sprouts ((
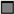
) NoPEF (
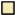
), 1.01 kV/cm and 2.7 kJ/kg, (
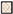
) 1.01 kV/cm and 27 kJ/kg, (
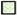
) 3.00 kV/cm and 2.8 kJ/kg, and (
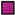
) 3.00 kV/cm and 27.7 kJ/kg). (**a**) Benzyl nitrile; (**b**) 2-butenenitrile; (**c**) 3-methyl butenenitrile; (**d**) 4-(methylthio) butanenitrile. Statistically significant differences are designated by different letters (*p* < 0.05, *n* = 6).

**Figure 6 foods-11-02892-f006:**
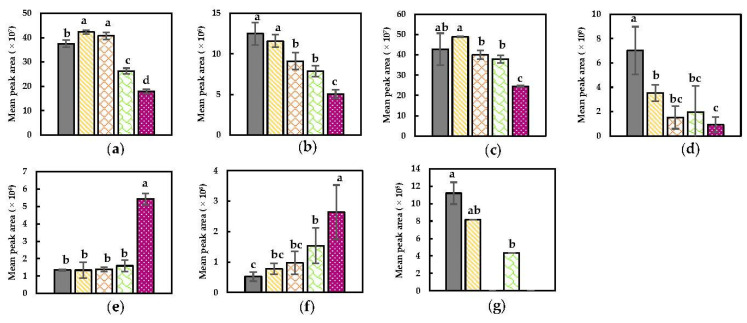
Specific compound plots of selected discriminant nitriles in the headspace of differently PEF-treated Brussels sprouts ((
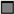
) NoPEF (
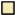
), 1.01 kV/cm and 2.7 kJ/kg, (
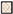
) 1.01 kV/cm and 27 kJ/kg, (
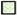
) 3.00 kV/cm and 2.8 kJ/kg, and (
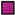
) 3.00 kV/cm and 27.7 kJ/kg). (**a**) Hexanal; (**b**) (E)-2-hexenal; (**c**) (Z)-2-hexenal; (**d**) 3-hexenal; (**e**) (Z)-3-hexen-1-ol; (**f**) (E,Z)-2,6-nonadienal; (**g**) (E,E)-2,4-heptadienal. Statistically significant differences are designated by different letters (*p* < 0.05, *n* = 6).

**Table 1 foods-11-02892-t001:** Parameter values for the different PEF treatments in each of the used treatment chambers.

	Medium–Sized Treatment Chamber		Small–Sized Treatment Chamber	
Dimensions of the electrodes (width × height × thickness)	20.0 × 20.5 × 0.5 cm		10.0 × 10.0 × 0.5 cm	
Interelectrode distance	29.7 cm		10.0 cm	
Volume of the treatment chamber	12.2 L		1.0 L	
Amount of sample inserted into the treatment chamber	650 g		188 g	
Amount of standardized tap water	4350 mL		300 mL	
Electrical field strength at 30 kV	1.01 kV/cm		3.00 kV/cm	
Pulse width	225 ± 19 µs ^a^		233 ± 6 µs ^a^	
Energy input per pulse	450 J/pulse		450 J/pulse	
Specific energy input per pulse	90 J/kg·pulse		922 J/kg·pulse	
(Total) specific energy input	2.7 kJ/kg (30 pulses)	27 kJ/kg (300 pulses)	2.8 kJ/kg (3 pulses)	27.7 kJ/kg (30 pulses)

^a^ Mean and standard error are based on 5 consecutive measurements (replicates).

**Table 2 foods-11-02892-t002:** VID, identity, chemical class and RI of markers (|VID| ≥ 0.800) for differently pretreated Brussels sprouts samples ^a^.

NoPT				
VID	Identity	Chemical Class	RI	Odor Description
0.991	Hexanal	Aldehyde	1096	Fatty, green, grassy, apple, rancid, tallow ^c,d,e,f,g^
0.987	(E)-2-Hexenal	Aldehyde	1232	Green, banana, fatty, apple, tallow ^c,f,g^
0.982	*Unidentified*	-	1480	
0.980	1,3-Pentadiene ^b^	Alkene	597	
0.979	(E,E)-2,4-Hexadienal	Aldehyde	1418	Green, fruity, citrus, waxy ^g^
0.977	*Unidentified*	-	1150	
0.959	(E,E)-2,4-Hexadienal	Aldehyde	1423	Green, fruity, citrus, waxy ^g^
0.953	(E,Z)-2,6-Nonadienal	Aldehyde	1599	
0.952	*Unidentified*	-	1422	
0.952	1-Penten-3-one	Ketone	1028	
0.950	Acetone	Ketone	819	
0.937	(E)-2-Nonenal	Aldehyde	1549	Rancid, cucumber, green, tallowy, fatty, oily ^g^
0.920	(E)-2-Pentenal	Aldehyde	1142	Strawberry, fruity, tomato, pungent, apple ^g^
0.874	8-Hydroxy-2,2-dimethyl-dec-5-en-3-one ^b^	Ketone	1977	
0.874	Cis-1,2-dimethyl cyclopropane ^b^	Cycloalkane	519	
0.873	2-Pentene	Alkene	523	
0.863	*Unidentified*	-	1155	
0.833	5-Ethyl-2(5H)-furanone	Furanic compound	1784	
0.828	Hexanenitrile	Nitrile	1315	
0.815	Methyl thiocyanate	Thiocyanate	1292	
−0.856	Pentanal	Aldehyde	984	Pungent ^e^
**Mix + Heat**				
**VID**	**Identity**	**Chemical Class**	**RI**	**Odor Description**
0.988	*Unidentified*	-	1727	
0.987	4-Ethyl benzaldehyde	Aldehyde	1731	
0.985	2-Ethyl furan	Furanic compound	957	Burnt, smoky, sweet ^c^
0.985	Dimethyl disulfide	Sulfurous compound	1088	Onion, sulfurous, cabbage, putrid, cauliflower, ripened cheese ^c,d,e,h,i,j^
0.981	Dimethyl trisulfide	Sulfurous compound	1404	Onion, sulfurous, alliaceous, fish, cabbage, spoiled, cooked, cauliflower ^c,d,e,f,h,j^
0.978	3-Hexen-1-ol	Alcohol	1401	Green, grassy ^j^
0.974	3-(Methylthio) nonanal ^b^	Aldehyde	1634	
0.971	2-Hexen-1-ol	Alcohol	1639	
0.963	2,2,6-Trimethyl-1-cyclohexene-1-carboxaldehyde	Aldehyde	1639	
0.955	1-Hexanol	Alcohol	1367	Resin, flower, green ^d^
0.948	Butanal	Aldehyde	881	Pungent ^e^
0.939	Cis-2-(2-pentenyl)furan	Furanic compound	1318	
0.937	Methyl (methylthio)methyl disulfide	Sulfurous compound	1685	
0.925	4-(Methylthio) butanenitrile	Nitrile	1813	
0.923	1-Penten-3-ol	Alcohol	1167	Buttery, pungent ^g^
0.898	1-Chloro pentane	Haloalkane	944	
0.890	(E)-3-Hexen-1-ol	Alcohol	1379	Green, grassy ^j^
0.889	5-Methyl-isoxazolidin-3-one ^b^	Ketone	1822	
0.867	2-Methyl-3-methylene-cyclopentanecarboxaldehyde ^b^	Aldehyde	1479	
0.846	2,4-Pentadienenitrile ^b^	Nitrile	1345	
0.841	4-Isothiocyanato-1-butene	Isothiocyanate	1474	
0.821	*Unidentified*	-	946	
0.821	2-Ethyl thiophene	Sulfurous compound	1186	
0.811	1-(Bicyclo [3.2.1]oct-2-en-4-yl)-4-phenyl-1,2,4-triazolidine-3,5-dione ^b^	Ketone	1781	
**PEF + Heat + Mix**				
**VID**	**Identity**	**Chemical Class**	**RI**	**Odor Description**
0.877	Pentane	Alkane	496	
0.823	2-Butenenitrile	Nitrile	1192	Pungent ^d^
−0.806	Methyl thiocyanate	Thiocyanate	1292	
−0.863	(E,E)-2,4-Heptadienal	Aldehyde	1506	Fatty, green, nut ^c,d,f,j^
−0.867	Cyano-3,4-epithiobutane ^b^	Nitrile/sulfurous compound	1973	
−0.878	(Z)-2-Penten-1-ol	Alcohol	1334	
−0.893	Isothiocyanato cyclopropane ^b^	Isothiocyanate/cycloalkane	1861	
−0.903	Benzyl nitrile	Nitrile	1955	Pickled, pungent ^f^
**Heat + Mix**				
**VID**	**Identity**	**Chemical Class**	**RI**	**Odor Description**
0.925	Pentane	Alkane	496	
0.922	2,4-Dimethyl hexane ^b^	Alkane	801	
0.911	2,2,4,6,6-Pentamethyl heptane	Alkane	956	
0.909	2-Butenenitrile	Nitrile	1192	Pungent ^d^
0.885	3-Methyl-2-butenenitrile ^b^	Nitrile	1289	
0.866	Ethyl cyclohexane	Cycloalkane	888	
0.827	Cis-1,3-dimethyl cyclohexane	Cycloalkane	839	
0.819	Pentanal	Aldehyde	984	Pungent ^e^
−0.811	Cyano-3,4-epithiobutane ^b^	Nitrile/sulfurous compound	1973	
−0.822	(E,E)-2,4-Heptadienal	Aldehyde	1506	Fatty, green, nut ^c,d,f,j^
−0.847	Isothiocyanato cyclopropane ^b^	Isothiocyanate/cycloalkane	1861	
−0.853	(Z)-2-Penten-1-ol	Alcohol	1334	
−0.890	Benzyl nitrile	Nitrile	1955	Pickled, pungent ^f^

^a^ Components, identified using the spectral library of NIST, that do not match with the RI found in literature, are indicated as ‘*unidentified’*. Components, for which the RIs are not found in literature are indicated as ‘*tentatively identified*’ (^b^). The components are listed in decreasing order of VID. A positive VID of a compound for a class conveys the presence of a higher concentration of that compound in that specific class compared to that compound in another class(es) whereas a negative VID denotes a lower concentration of that compound in that specific class. If found in the literature, the odor description of the marker is added ([2] ^e^; [20] ^h^; [23] ^c^; [62] ^d^; [63] ^i^; [64] ^j^; [65] ^f^; [66] ^g^).

**Table 3 foods-11-02892-t003:** VID, identity, chemical class and RI of markers (|VID| ≥ 0.800) for differently PEF-treated Brussels sprouts ^a^.

NoPEF				
VID	Identity	Chemical Class	RI	Odor Description
0.955	Ethyl cyclohexane	Cycloalkane	886	
0.948	Cis-1,3-dimethyl cyclohexane	Cycloalkane	815	
0.908	2,2,4,6,6-Pentamethyl heptane	Alkane	954	
0.897	1,3-Pentadiene ^b^	Alkene	596	
0.874	3-Ethyl-1,5-octadiene	Alkene	1029	
0.872	Vinyl crotonate ^b^	Ester	1045	
0.859	Trans-1,2-dimethyl cyclohexane ^b^	Cycloalkane	838	
0.835	3-Hexenal	Aldehyde	1155	Green, leafy, fruity, apple-like ^c^
0.834	5,5-Dimethyl-1,3-hexadiene ^b^	Alkene	898	
0.828	2,2,4-Trimethyl pentane	Alkane	709	
0.826	(E,E)-2,4-Heptadienal	Aldehyde	1479	Fatty, green, nut ^c,d,e,f^
0.818	2,4-Dimethyl hexane	Alkane	799	
0.803	3-Ethyl-1,5-octadiene	Alkene	1013	
**1.01 kV/cm and 2.7 kJ/kg**				
**VID**	**Identity**	**Chemical Class**	**RI**	**Odor Description**
−0.806	4,4-Dimethyl-3-oxopentanenitrile ^b^	Nitrile	1247	
−0.820	Tetrahydrofuran	Furanic compound	861	
−0.906	Benzyl nitrile	Nitrile	1954	Pickled, pungent ^f^
**1.01 kV/cm and 27.0 kJ/kg**				
**VID**	**Identity**	**Chemical Class**	**RI**	**Odor Description**
0.946	2,4-Dimethyl-1-heptene	Alkene	881	
0.936	4-Methyl heptane	Alkane	720	
0.869	2-Methyl-3-methylene cyclopentanecarboxaldehyde ^b^	Aldehyde	1479	
0.856	2-Methyl-3-methylene cyclopentanecarboxaldehyde ^b^	Aldehyde	1432	
0.815	1-Nonen-4-ol ^b^	Alcohol	1662	
−0.873	Hexanenitrile	Nitrile	1315	
**3.00 kV/cm and 2.8 kJ/kg**				
**VID**	**Identity**	**Chemical Class**	**RI**	**Odor Description**
0.897	Ethyl acetate	Ester	893	Fruity, brandy-like, pineapple ^c^
0.812	4-(Methylthio) butanenitrile	Nitrile	1812	
**3.00 kV/cm and 27.7 kJ/kg**				
**VID**	**Identity**	**Chemical Class**	**RI**	**Odor Description**
0.969	2-Butenenitrile	Nitrile	1192	Pungent ^d^
0.953	(Z)-3-Hexen-1-ol	Alcohol	1399	Green, grass ^c,g^
0.898	3-Methyl butanenitrile	Nitrile	1139	
0.886	4,4-Dimethyl-3-oxopentanenitrile ^b^	Nitrile	1247	
0.876	(E,Z)-2,6-Nonadienal	Aldehyde	1598	
0.837	Benzyl nitrile	Nitrile	1954	Pickled, pungent ^f^
0.829	Tetrahydrofuran	Furanic compound	861	
0.825	*Unidentified*	-	1289	
−0.888	(Z)-2-Hexenal	Aldehyde	1230	
−0.891	Hexanal	Aldehyde	1095	Fatty, green, grassy, apple, rancid, tallow ^c,d,f,h,i^
−0.909	(E)-2-Hexenal	Aldehyde	1212	Green, banana, fatty, apple, tallow ^c,f,i^

^a^ Components, identified using the spectral library of NIST, which are not matching with the RI found in literature, are indicated as ‘*unidentified’*. Components, identified using the spectral library of NIST, for which the RIs are not found in literature are indicated as ‘*tentatively identified*’ (^b^). The components are listed in decreasing order of VID. A positive VID of a compound for a class conveys the presence of a higher concentration of that compound in that specific class compared to that compound in another class(es) whereas a negative VID denotes a lower concentration of that compound in that specific class. If found in the literature, the odor description of the marker is added ([2] ^h^; [20] ^g^; [23] ^c^; [62] ^d^; [64] ^e^; [65] ^f^; [66] ^i^).

## Data Availability

Data are contained within the article.

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
