# Peer review of "The Volatile Profile of Brussels Sprouts (Brassica oleracea Var. gemmifera) as Affected by Pulsed Electric Fields in Comparison to Other Pretreatments, Selected to Steer (Bio)Chemical Reactions"

_foods, 2022, doi:10.3390/foods11182892_

Round 1
Reviewer 1 Report
Dear authors and editors’ thanks for your work and give me chance to read it. Kindly check the following comments.
Manuscript format: First of all the manuscript need to revise according to the journal format; from the title till the references.
Abstract: it is not well-written and should be completely revised. It should reflect the entire contents of the manuscript, in abbreviated form, including: Background and problem, the rationale for the study; research objectives; some methodology; important data including statistical analysis; conclusions, novelty, the importance of the findings.
Line 143, add the reference
Do you have two raw materials from different resources? Please check, lines 123, and 194, if yes, how you compare the results. All the experiments should be done on the same material.
Line 224, please check the size of the amber glass vial, 25 ml?
Line 225, why you added NaCl, please mention the reason and add the reference.
Line 253, what about the purity of the Helium gas? please check and add
What about the running time of each sample, how long? Please check and add
What about the scanning duration of the mass spectra? Please check and add
For tables 2 and 3, I suggested authors add the available odor description (one column) of the volatile Components
Reviewer 2 Report
The article entitled “the volatile profile of Brussels sprouts (Brassica oleracea var. gemmifera) as affected by pulsed electric fields in comparison to other pretreatments, selected to steer (bio)chemical reactions” was written. The article showed interesting results and few comments for minor revisions as follows
1. Line 599-605 had a long sentence It is better to separate in two sentences
2. In line 614 tp 615, Author mentioned that the incubation at 40 oC played important role on nitriles. It is not clear what the roles of the incubation. If it had an effect on induction and degradation of compounds, the sentence needs a confirmation from other studies related to this statement.
